# The Expanding Phenotype of ZTTK Syndrome Due to the Heterozygous Variant of *SON* Gene Focusing on Liver Involvement: Patient Report and Literature Review

**DOI:** 10.3390/genes14030739

**Published:** 2023-03-17

**Authors:** Andrea Pietrobattista, Luca Della Volpe, Paola Francalanci, Lorenzo Figà Talamanca, Lidia Monti, Francesca Romana Lepri, Maria Sole Basso, Daniela Liccardo, Claudia Della Corte, Antonella Mosca, Tommaso Alterio, Silvio Veraldi, Francesco Callea, Antonio Novelli, Giuseppe Maggiore

**Affiliations:** 1Hepatogastroenterology, Nutrition, Digestive Endoscopy and Liver Transplant Unit, Bambino Gesù Children’s Hospital, IRCCS, 00165 Rome, Italy; 2Pathology Unit, Department of Diagnostic and Laboratory Medicine, Bambino Gesù Children’s Hospital, IRCCS, 00165 Rome, Italy; 3Department of Radiology, Bambino Gesù Children’s Hospital, IRCCS, 00165 Rome, Italy; 4Laboratory of Medical Genetics, Translational Cytogenomics Research Unity, Bambino Gesù Children’s Hospital, IRCCS, 00165 Rome, Italy; 5Department of Histopathology, Bugando Medical Centre, Catholic University of Healthy Allied Sciences, Mwanza P.O. Box 1370, Tanzania

**Keywords:** chronic liver disease, ZTTK syndrome, *SON* mutation, brain malformations, developmental delay

## Abstract

Zhu–Tokita–Takenouchi–Kim (ZTTK) syndrome, an intellectual disability syndrome first described in 2016, is caused by heterozygous loss-of-function variants in *SON*. Haploinsufficiency in *SON* may affect multiple genes, including those involved in the development and metabolism of multiple organs. Considering the broad spectrum of *SON* functions, it is to be expected that pathogenic variants in this gene can cause a wide spectrum of clinical symptoms. We present an additional ZTTK syndrome case due to a de novo heterozygous variant in the *SON* gene (*c.5751_5754delAGTT*). The clinical manifestations of our patient were similar to those present in previously reported cases; however, the diagnosis of ZTTK syndrome was delayed for a long time and was carried out during the diagnostic work-up of significant chronic liver disease (CLD). CLD has not yet been reported in any series; therefore, our report provides new information on this rare condition and suggests the expansion of the ZTTK syndrome phenotype, including possible liver involvement. Correspondingly, we recommend screening patients with *SON* variants specifically for liver involvement from the first years of life. Once the CLD has been diagnosed, an appropriate follow-up is mandatory, especially considering the role of *SON* as an emerging player in cancer development. Further studies are needed to investigate the role of *SON* haploinsufficiency as a downregulator of essential genes, thus potentially impairing the normal development and/or functions of multiple organs.

## 1. Introduction

Zhu–Tokita–Takenouchi–Kim (ZTTK, OMIM#617140) syndrome is an autosomal dominant hereditary disease caused by heterozygous variants in the *SON* gene (OMIM#182465, GenBank#NC_000021.9) located on chromosome 21q22.11 [1,2,3]. *SON* plays an important role in cell cycle progression and affects RNA splicing as a splicing cofactor [3,4]. The haploinsufficiency of this gene can lead to intron retention and exon skipping, especially at weak splice sites, which affects multiple genes, including various genes involved in organ development and metabolism [5,6]. *SON* is also involved in the pluripotency and survival of embryonic stem cells, as well as in the alternative splicing of other genes involved in epigenetic regulation and apoptosis [3,5,7,8]. Furthermore, *SON* has been shown to be involved in the transcriptional regulation of genes associated with carcinogenesis [5]. The ZTTK phenotypic spectrum and the pathogenicity of missense variants have not been evaluated in detail. Considering the broad spectrum of *SON* functions, the pathogenic variants in this gene are expected to cause diverse clinical symptoms [9]. Reported clinical characteristics mainly include developmental delay, brain malformations, facial dysmorphisms, eye and/or vision abnormalities, urogenital malformations, and craniosynostosis [1,2,3,9,10]. Here, we report an additional ZTTK syndrome case in light of a de novo heterozygous variant in *SON*, detected during an extensive diagnostic work-up of chronic liver disease (CLD). The clinical manifestations of our patient were similar to those present in previously reported cases and series, but to our knowledge, this is the first case report that identifies the importance of liver involvement in ZTTK syndrome. This report provides new information on this rare new condition.

## 2. Patients and Methods

### 2.1. Case Description

The proband, a 16-year-old male, is the first child of healthy non-consanguineous parents whose family history did not indicate the presence of any genetic diseases. He was born at term after an uneventful pregnancy with a normal neonatal period. At 9 months of age, a developmental delay became evident; thus, CGH array testing was performed with unspecific results, while extended endocrine–metabolic screening revealed a primary hypothyroidism, initiating levothyroxine. Psycho-motor rehabilitation was then conducted and maintained in the years that followed, with a learning cognitive support system at school that ensured autonomous walking at 16 months, spontaneous language at around 3 years, and sphincter control around 4 years. At 5 years of age, he underwent two surgical corrections of bilateral vesicoureteral reflux.

When the patient was 12, a routine abdominal ultrasound, performed as part of his urogenital malformation follow-up, showed hyperechoic and heterogeneous liver parenchyma. At that time, liver function tests (LFTs) were all within the normal range, and therefore, no action was taken. However, 4 years later, a follow-up ultrasound revealed increased liver abnormalities, which now included all signs of advanced chronic liver involvement. At this stage, he was referred to our Pediatric Hepatology Center for a full review. At the time of admission, the patient was in good general condition and was asymptomatic, with a body mass index of 16.6 (<3°). At clinical examination, he presented a long-limbed habitus and ligamentous hyperlaxity, but not muscle hypotonia, arachnodactyly of the fingers, and bilateral flat foot, as well as no alteration in the skin or nails. Mild splenomegaly was the only sign of CLD.

Facial dysmorphisms were noted, including a hairline low on the forehead; horizontalized thick eyebrows; downward eyelids; low-set extroverted ears; a thin, arched upper lip; a fleshy and extroverted lower lip; small teeth; and dental diastemas.

An extensive diagnostic work-up of CLD was performed with laboratory, radiological, and clinical investigations, which finally ruled out the most frequent and known causes of CLD at that age. LFTs were particularly unremarkable with no abnormal values of cholestasis, liver enzymes, tumor markers, and hepatic synthetic function. A full blood count revealed mild thrombocytopenia due to hypersplenism secondary to the enlarged spleen size. An abdominal doppler ultrasound scan and magnetic resonance imaging (MRI) confirmed the chronic liver involvement, showing hepatic parenchymal micro- and macronodular transformation, as well as splenomegaly (Figure 1). Upper gastro-intestinal endoscopy ruled out esophageal varices but showed mild portal hypertension (PH)-related gastropathy. In addition, we performed echocardiography and ophthalmology reviews, which were both normal, while a psychological evaluation was performed, revealing a non-verbal IQ of 63 (which was in keeping with mild disability) through the Raven matrices.

A liver biopsy was then performed to further investigate CLD etiology. Histology revealed micro- and macronodular cirrhosis characterized by disturbed architecture due to the presence of veno-venous fibrotic septa that delimited parenchymal nodules (Figure 2). A slight inflammatory lymphomonocyte infiltrate was present in the septa and in the portal spaces. Close to an anatomical septum, a cluster of optically empty cells intermixed with atrophic hepatocytes was present (Figure 3). These cells, found with immunohistochemical analysis, were hepatocyte paraffin 1 (HepPar1)-, smooth muscle actine (SMA)+, desmin-, glial fibrillary acidic protein (GFAP)-, and S100-, and they were consistent with Ito perisinusoidal cells (PSCs) in a setting of s.c. spongiotic pericytoma. The absence of a conclusive liver diagnosis and multiorgan involvement, suggestive of a genetic syndrome, made it mandatory to perform a genetic test; this was at first examined with a targeted liver genes panel (Appendix A) and then expanded to whole-exome sequencing (WES). The genetic test showed a heterozygous de novo frameshift variant *c.5751_5754delAGTT* in the *SON* gene. This variant is classified as pathogenetic (class 5), according to ACMG guidelines (Figure 4), and is already reported in the medical literature [9]. The finding of the *SON* mutation was unexpected from a hepatology point of view, advocating an overall patient phenotype reassessment, which was ultimately found to be highly compatible with ZTTK syndrome, according to the literature (Table 1) [2,3,9,10,11,12,13,14,15,16,17]. To further investigate the possible link with ZTTK syndrome, a brain MRI was performed, and its findings were highly similar to previous descriptions (Figure 1).

The case study is now under a regular follow-up for CLD, especially monitoring PH evolution and eventual hepatic focal lesion development.

### 2.2. Whole-Exome Sequencing (WES)

Clinical exome sequencing, using standard procedures, was performed on genomic DNA extracted from the circulating leukocytes of the proband and his parents. Library preparation was carried out in accordance with the manufacturer’s protocol (Twist Bioscience HQ, South San Francisco, CA, USA) and sequenced on a NovaSeq6000 (Illumina, San Diego, CA, USA) platform. The target parameters were the coding exons, including a region extension of 25 bases from the 3′ end and 25 bases from the 5′ end (based on the RefSeq database). We obtained a targeted next-generation sequencing (NGS) assay with a mean 150× coverage for 97% bases, a specificity of 100%, and a sensitivity of 100%, as well as a quality score of ≥30. The BaseSpace pipeline and TGex software LifeMap Sciences (https://www.lifemapsc.com/, accessed on 10 February 2023) were used to call and annotate variants, respectively. Sequencing data were aligned to the hg19 human reference genome. Based on the guidelines of the American College of Medical Genetics and Genomics (ACMG), a minimum depth coverage of 30× was considered suitable for analysis [18]. Variant calling was performed with the Dragen Germline Enrichment application of BaseSpace (Illumina, San Diego, CA, USA), while variant annotation and the phenotype-based prioritization of candidate genes were carried out using Geneyx Analysis software (Geneyx Genomex, http://www.geneyx.com, accessed on 10 February 2023).

## 3. Discussion

Since the first case described by Zhu et al. in 2016, a limited series of patients with the *SON* gene mutation were reported with recurrent phenotypic characteristics that together help to define ZTTK syndrome. Interestingly, our patient presented most of the reported manifestations, but in addition, he presented a yet unreported liver pathology consisting of liver cirrhosis. Moreover, a peculiar elementary lesion as spongiosis hepatis or spongiotic pericytoma was evident. Spongiosis hepatis is rather unusual in humans, and it has been shown to derive from the transformation of PSCs in spongiotic cells [19]. The biologic meaning of spongiosis hepatitis and spongiotic pericytoma is still under debate; however, it is known that PSCs are the key cells in hepatic fibrogenesis and the cirrhotic septa formation through their transformation into myofibroblasts and fibroblasts [20].

The diagnosis of ZTTK syndrome while CLD was being investigated was utterly unexpected, but the finding could be embedded and interpreted in the new paradigm of genetic spectra for inherited pediatric liver diseases.

CLDs often have underlying genetic disorders that not only embrace liver-based diseases but also systemic diseases. These conditions are often present with overlapping phenotypes and lack specific laboratory biomarkers, which may lead to delays in diagnoses, improper treatment, and new associations. As a result, the incidence of monogenic liver diseases is not known, but nearly half of the CLDs that present themselves in childhood have a genetic basis [21], thus supporting the expanded use of WES analysis for genetic tests compared to liver-based panels.

Liver disease in our patient clearly shows a chronic fibrosis with micro-macronodularity, which reflects the parenchymal disturbed architecture seen in histology. Interestingly, biochemical investigations have always revealed normal LFTs, even in the last follow-up review, thus lacking any correlation with the progression of CLD. Normal LFTs could be a pitfall in both investigating and excluding liver diseases. Therefore, the normal LFTs reported in all previously published single cases and series of ZTTK syndrome may not be completely reassuring. Moreover, a pattern of normal biochemical tests in CLDs is not completely unexpected because there are other known disease models which can progress toward PH maintaining normal LFTs. In our case, liver involvement was accidentally detected at age 16 during a follow-up ultrasound that monitored the surgical vesicoureteral reflux correction. A proper ecographic evaluation of the liver requires dedicated pediatric radiological skills as well as a doppler study of the hepatic vascular flows which could be missed during routine studies.

The role of *SON* as a master regulator of genes that are essential for human neurodevelopmental processes, kidney development, and metabolism has already been revealed through a description of various essential genes that are significantly downregulated upon *SON* haploinsufficiency, thus potentially impairing normal development and/or functions of multiple organs [3,4]. As such, it is important to identify direct *SON* targets in cells that are relevant to the observed clinical phenotypes.

Overall, it is still not clear whether hepatic involvement is a possible phenotypic manifestation associated with a mutation of the *SON* gene or if it is an epiphenomenon in the context of patients with ZTTK syndrome. However, in the absence of other definitive causes, it seems reasonable to include liver disease in a proposal of expanding phenotypes.

Unfortunately, in our case, even if we excluded variants in all genes known to be associated with CLD pathogenesis, we would not be able to investigate whether there was a downregulation of those genes due to *SON* haploinsufficiency. In this view, the absence of functional studies is the major limitation of our study that prevents us from establishing a direct link between the *SON* variant and the CLD. At this stage, we can only surmise that this hypothesis is worthy of further consideration as a field of future research mainly focusing on *SON* and mechanisms of hepatic fibrogenesis.

Interestingly, a possible link between a Hepatitis B virus (HBV) infection and the *SON* gene has been previously investigated in a study by Sun et al., who found that the transcription repression of human HBV genes is inhibited by negative regulatory binding protein *SON* [22]. This potential pathway should be taken into consideration as a mechanism driving CLD, and clinicians involved in this rare syndrome should be aware of this association.

It is also noteworthy that *SON* is recognized as an emerging player in cancer development and progression due to the multiple *SON* target genes which are directly involved in cell proliferation and genome stability. This aspect results in a significant adjunctive risk factor in patients with macronodular CLD, which independently initiates the risk of carcinogenesis and warrants a proper follow-up.

In conclusion, we recommend a specific screening test for liver involvement in all patients with ZTTK syndrome. Given the possibility of normal LFTs in the context of liver fibrosis, the execution of serial doppler ultrasound scans with a focus on CLD signs is advocated, which may also help clinicians to better recognize, properly screen, and effectively manage this novel and rare condition.

## Figures and Tables

**Figure 1 genes-14-00739-f001:**
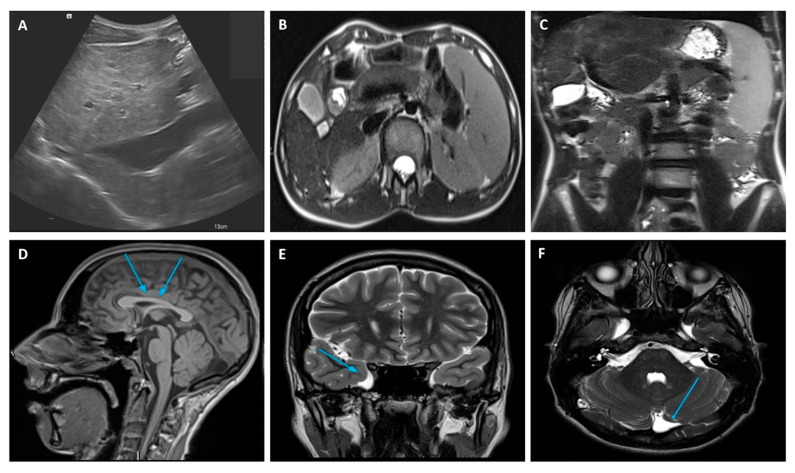
Liver doppler ultrasound (**A**) and abdominal MRI (**B**) findings showing hepatic parenchymal micro- and macronodular transformation with associated splenomegaly (**C**). MRI brain abnormalities: (**D**) mild hypoplasia of the rostrum and splenium of the corpus callosum, (**E**) temporo-polar arachnoid cyst, (**F**) retro-infero-cerebellar arachnoid cyst. MRI, Magnetic Resonance Imaging.

**Figure 2 genes-14-00739-f002:**
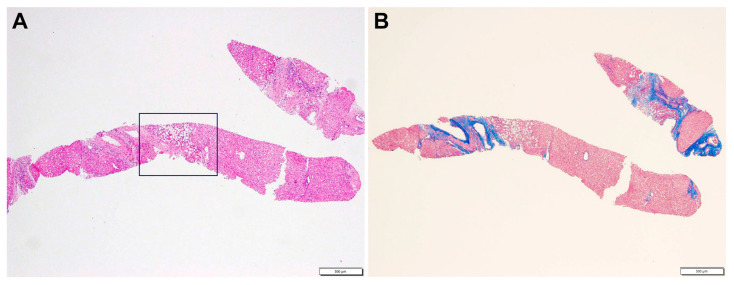
Liver biopsy: (**A**) expanded portal tract with mild inflammatory infiltrate. Inset displays a cluster of empty cells near an anatomical septum (HE, 4×). (**B**) Disturbed architecture due to fibrous porto-portal bridges consistent with cirrhosis (Masson 4×).

**Figure 3 genes-14-00739-f003:**
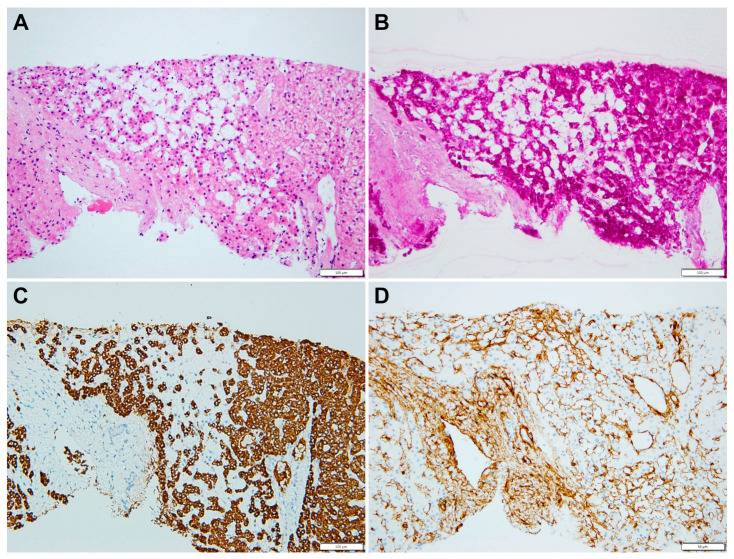
(**A**) Empty cells are mixed with small (atrophic) hepatocytes (HE, 20×). (**B**) Empty cells have no glycogen in the cytoplasm (PAS, 20×); (**C**) Empty cells are not abnormal hepatocytes (HepPar1 negative, 20×) and (**D**) they are surrounded by smooth muscle actin filaments (SMA, 20).

**Figure 4 genes-14-00739-f004:**
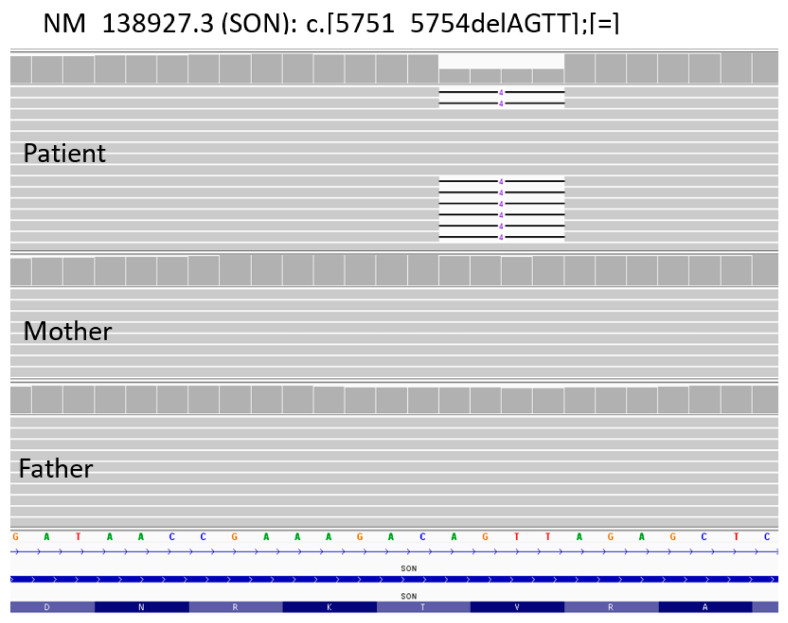
NGS sequencing data from blood samples of the proband and his parents. Identification of de novo variant *c.5751_5754delAGTT* (p.Val1918fsTer87) in *SON* gene.

**Table 1 genes-14-00739-t001:** Clinical characteristics of previously reported patients with ZTTK syndrome.

Clinical Findings	Current Case	Takenouchi et al. [10]	Kim et al. [3]	Tokita et al. [2]	Yang et al. [13]	Slezak et al. [14]	Quintana Castanedo et al. [15]	Tan et al. [16]	Yang et al. [17]	Dingemans et al. [9]
Chronic liver disease	+									
Developmental and/or intellectual disability	+	+	+	+	+	+	+	+	+	+
Seizures		+	+	+		+		+	+	+
Hypotonia			+	+		+	+	+		+
Abnormal brain imaging	+		+	+	+	+	+	+		+
Facial dysmorphisms	+	+	+	+	+	+	+	+		+
Growth delay		+	+	+	+		+	+	+	+
Eye/vision abnormality			+	+		+	+			+
Congenital heart defects		+	+	+		+	+			+
Feeding difficulties	+					+				+
Gastrointestinal malformation			+						+	+
Genitourinary anomalies	+		+	+		+				+
Musculoskeletal abnormalities	+		+	+	+	+				+

## Data Availability

The data presented in this study are available on request from the corresponding author.

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
