# Peer review of "The Expanding Phenotype of ZTTK Syndrome Due to the Heterozygous Variant of *SON* Gene Focusing on Liver Involvement: Patient Report and Literature Review"

_genes, 2023, doi:10.3390/genes14030739_

Round 1
Reviewer 1 Report
This manuscript is a case report of only one male patient of ZTTK syndrome, presenting with a significant chronic disease, which has not been reported in previous reports.
From an educational point, this report can provide new information on this rare syndrome. There seems no needs of revisions, although I CANNOT find any figures, or Figure 1, 2, 1nd 3, in the manuscript.
Author Response
This manuscript is a case report of only one male patient of ZTTK syndrome, presenting with a significant chronic disease, which has not been reported in previous reports.
From an educational point, this report can provide new information on this rare syndrome. There seems no needs of revisions, although I CANNOT find any figures, or Figure 1, 2, 1nd 3, in the manuscript.
Response: We thank the reviewer for his positive evaluation. We are sorry he was not able to see the figures which clearly show the severe degree of the unexpected liver involvement (radiology and histology) and the brain MRI features of ZTTK syndrome. Tabs and figures were uploaded in a zip file as requested by journal but we would be happy to attach them here to reviewer if he is not able to open the file.
Reviewer 2 Report
In this manuscript, Pietrobattista and colleagues describe the clinical presentation of a ZTTK syndrome case due to a de novo heterozygous variant in the SON gene. Although syndromic signs of the clinical manifestation were common to the ones presented by ZTKK individuals described for this condition, in this case the ZTTK diagnosis was not detected in early life but carried out during diagnostic work-up of a significant chronic liver disease (CLD) in the patient. The study is new in suggesting CLD or even more broad liver involvement in ZTTK studies; suggesting a rare condition that expands on the currently known syndromic phenotype. Authors recommend screening patients with SON variants specifically for liver involvement from early detection and diagnosis of ZTTK. According to authors the significance of the study resides in the role of SON as an emerging player in cancer development and conclude that further studies are needed to investigate the role of SON haploinsufficiency as a downregulator of essential genes, thus potentially impairing the normal development and/or functions of multiple organs.
The study is well presented and the design and execution are clinically sound. However, to this reviewer’s eyes, the manuscript lacks of exhaustive literature review when drawing main conclusions. After a brief literature review looking into the possible link of SON gene and Liver function, the fact that mutations in this gene result in abnormal liver biology is not surprising (see below), yet seeing this to translate into CLD is a great confirmation and advancement in the field.
Major concern:
The study and its main conclusion can benefit from including current knowledge on the role of SON in the Liver, specifically in Hepatitis B virus (HBV) infection, where transcription repression of human HBV genes is inhibited by negative regulatory binding protein SON. Thus, the discussion section would benefit from acknowledging this SON-HBV link, so the field is aware of potential complications in SON mutation carriers when diagnoses with HBV. This link should also be considered for more general Liver disease outcomes in ZTTK SON mutation carriers, and for looking into potential pathways driving the described CLD. Finally the authors can acknowledge patient is not carrier of HBV, assuming they have the ability to confirm this. Relevant literature to review: PMID: 11306577, 33042753.
This reviewer would be happy to review a revised version of the manuscript.
Author Response
Major concern:
The study and its main conclusion can benefit from including current knowledge on the role of SON in the Liver, specifically in Hepatitis B virus (HBV) infection, where transcription repression of human HBV genes is inhibited by negative regulatory binding protein SON. Thus, the discussion section would benefit from acknowledging this SON-HBV link, so the field is aware of potential complications in SON mutation carriers when diagnoses with HBV. This link should also be considered for more general Liver disease outcomes in ZTTK SON mutation carriers, and for looking into potential pathways driving the described CLD. Finally the authors can acknowledge patient is not carrier of HBV, assuming they have the ability to confirm this. Relevant literature to review: PMID: 11306577, 33042753.
This reviewer would be happy to review a revised version of the manuscript.
Response: We thank the reviewer to bring up this point. We agreed that a SON-HBV link is noteworthy to be highlighted to let aware the clinical community involved in this rare syndrome of the possible complication of HBV infection. This experimental study form SUN et al of the 2001 was not repeated in literature but it suggest potential field for future research on CLD in SON mutation carriers. Moreover, this SON-HBV link seems relevant considering the role of SON as an emerging player in cancer development as stated in the manuscript. We added a paragraph to the discussion and the study by Sun et al (PMID 11306577) in the references list.
Our patient was fully vaccinated for HBV as for the immunization schedule in use in our country. Moreover neither the always normal liver enzymes nor the liver histology without any signs of chronic infection were indication for further evaluation.
Reviewer 3 Report
Pietrobattista and colleagues have exposed a possible expansion of ZTTK syndrome with the description of a SON heterozygous variant. Their manuscript could be interesting, but it suffers from several issues.
1. There is no information about the filtering strategy for exome analysis. We may suppose that other candidates could be obtained for the patient, moreover regarding the hepatic phenotype. A table with the description of other candidates must be provided.
2. They have described a de novo frameshift c.5751_5754delAGTT variant. We haven’t any information about the transcript, or the genomic coordinates. By having a look in variant validator tool and mutalyzer, it seems that the variant should be described as NM_138927.4:c.5753_5756del p.(Val1918GlufsTer87), and not as c.5751_5754delAGTT. In that case, the author would identify the recurrent 4-bp deletion observed in the ZTTK syndrome, and well described in Kushary et al, 2021, and in the nice review from Dingemans et al, 2021, which was cited by the authors (N°9). The authors must change the coordinates and referred their variant as the recurrent and common variant observed in ZTTK syndrome.
3. The authors have exposed their hypothesis about the link between the chronic liver disease and the presence of the SON variant. An exploration of liver function was also conducted by Yang and colleagues (PMID 31557424), not cited here, with a normal liver function/normal doppler evaluation. There is also no description of CLD in the description of 52 patients, including several patients with the same variant reported here. As the authors didn’t perform any functional analysis in order to establish a link between SON variant and CLD, they must expose more clearly their hypothesis in the discussion. Unfortunately, there is a lack of proof and evidence in the discussion to back up this assumption.
Author Response
- There is no information about the filtering strategy for exome analysis. We may suppose that other candidates could be obtained for the patient, moreover regarding the hepatic phenotype. A table with the description of other candidates must be provided.
Response: We thank the reviewer to bring up this point. We explained in the manuscript that we started filtering the genetic test with an extended targeted genes panel for known liver diseases and then we expanded the research outside the liver phenotype. We understand that adding a table with the list of genes included in the panel (see attached) could have been useful and thus we proposed to upload it as supplemental (S2).
- They have described a de novo frameshift c.5751_5754delAGTT variant. We haven’t any information about the transcript, or the genomic coordinates. By having a look in variant validator tool and mutalyzer, it seems that the variant should be described as NM_138927.4:c.5753_5756del p.(Val1918GlufsTer87), and not as c.5751_5754delAGTT. In that case, the author would identify the recurrent 4-bp deletion observed in the ZTTK syndrome, and well described in Kushary et al, 2021, and in the nice review from Dingemans et al, 2021, which was cited by the authors (N°9). The authors must change the coordinates and referred their variant as the recurrent and common variant observed in ZTTK syndrome.
Response: According to our NGS data the genomic location is Chr21:34927288 - 34927291 (on Assembly GRCh37), at position g.34927288 is located and Adenine the first nucleotide deleted as shown on Fig.S1 (here attached). The identified deletion at nucleotide level corresponds to NM_138927: c.5751_5754delAGTT and at protein level correspond to p.Val1918GlufsTer87. We agree with the reviewer that is the same mutation because the effect on the protein doesn’t change. In the manuscript we describe as already known (Ref.PMID: 27545680), and annotated in the ClinVar database (ID: 252929) based on the protein change. We know that is variation is always reported as NM_138927.4:c.5753_5756del , but we prefer to describe exactly our NGS data. Following the reviewer’s suggestion, we will add in the manuscript also the description of the variation at protein level.
The authors have exposed their hypothesis about the link between the chronic liver disease and the presence of the SON variant. An exploration of liver function was also conducted by Yang and colleagues (PMID 31557424), not cited here, with a normal liver function/normal doppler evaluation. There is also no description of CLD in the description of 52 patients, including several patients with the same variant reported here. As the authors didn’t perform any functional analysis in order to establish a link between SON variant and CLD, they must expose more clearly their hypothesis in the discussion. Unfortunately, there is a lack of proof and evidence in the discussion to back up this assumption.
Response: We appreciate this observation form the reviewer. Indeed, liver disease in not present in any patient among the reported cases. In this light, there was Table 1 that summarized all the clinical studies present in literature including the one from Yang et al which is the study reference 13. Other reviewers struggled to have access to the zip file with table and figures therefore we attached table 1 to this revision. Some of studies reported in literature referred about normal liver tests and ultrasound, however we advised that also our patient had normal liver investigations until a dedicated hepatology work-up happened. CLD might be “mute” from a clinical point of view and often displaying normal blood tests even at cirrhotic stage. Liver ultrasound in children may require specific expertise to detect signs of CLD which are more evident on MRI.
We recognized in the manuscript that a limitation of our study is the lack of functional analysis to make a proof about the link between SON and CLD. At this stage we can only make a hypothesis based on this new association but with a well-documented message so the field is aware of potential complications in SON mutation carriers. Of course, this will be clearly the focus for further research, and it represents our intention.
We can stress this point in the discussion if it is the reviewer advice.

Round 2
Reviewer 2 Report
Authors have revised the manuscript to this reviewer's satisfaction. The addition hassufficiently improved the manuscript. I consider this work important for the field and should warrant publication in Genes. Thank you.
Author Response
I consider this work important for the field and should warrant publication in Genes. Thank you.
Response: We thank the reviewer, his help lead to a better manuscript.
Reviewer 3 Report
A modification of figure 4 should be assessed. There are irrelevant characters.
Author Response
A modification of figure 4 should be assessed. There are irrelevant characters.
Response: we thank the reviewer for his suggestion. please find attached a new version of figure 4.